# Increased CD4^+^CD8^+^ Double Positive T Cells during Hantaan Virus Infection

**DOI:** 10.3390/v14102243

**Published:** 2022-10-13

**Authors:** Huiyuan Zhang, Yazhen Wang, Ying Ma, Kang Tang, Chunmei Zhang, Meng Wang, Xiyue Zhang, Manling Xue, Xiaozhou Jia, Haifeng Hu, Na Li, Ran Zhuang, Boquan Jin, Lihua Chen, Yun Zhang, Yusi Zhang

**Affiliations:** 1Department of Immunology, School of Basic Medicine, Fourth Military Medical University, Xi’an 710032, China; 2Department of Immunology, School of Basic Medical Sciences, Yan’an University, Yan’an 716000, China; 3Department of Pathogenic Biology, School of Basic Medical Sciences, Yan’an University, Yan’an 716000, China; 4Eighth Hospital of Xi’an, Xi’an 710061, China; 5Center for Infectious Diseases, Tangdu Hospital, Fourth Military Medical University, Xi’an 710038, China; 6Department of Transfusion Medicine, Xijing Hospital, Fourth Military Medical University, Xi’an 710032, China

**Keywords:** HTNV, HFRS, DP T cells, CD8

## Abstract

Hantaan virus (HTNV) infection causes an epidemic of hemorrhagic fever with renal syndrome (HFRS) mainly in Asia. It is well known that T cells mediated anti-viral immune response. Although previous studies showed that double positive T (DP T) cells, a little portion of T lymphocytes, were involved in adaptive immune response during virus infection, their kinetic changes and roles in HTNV infection have not yet been explored. In this study, we characterized DP T cells from HFRS patients based on flow cytometry data combined with scRNA-seq data. We showed that HTNV infection caused the upregulation of DP T cells in the peripheral blood, which were correlated with disease stage. The scRNA-seq data clustered DP T cells, unraveled their gene expression profile, and estimated the ordering of these cells. The production of granzyme B and CD107a from DP T cells and the abundant TCR distribution indicated the anti-viral property of DP T cells. In conclusion, this study identified, for the first time, an accumulation of DP T cells in the peripheral blood of HFRS patients and suggested these DP T cells belonging to CD8^+^T cells lineage. The DP T cells shared the similar characteristics with cytotoxic T cells (CTL) and exerted an anti-viral role in HFRS.

## 1. Introduction

Hemorrhagic fever with renal syndrome (HFRS) is an acute infectious disease caused by Hantaan virus (HTNV) infection. About 90% of the HFRS cases worldwide have been reported in China, with the mortality rate up to 15% [1,2]. In Shaanxi Province, the HFRS incidence continues to be significantly higher than the national average [3]. Fever, hemorrhage and acute renal failure are the major syndromes of HFRS [4]. The typical clinical process containing five stages (fever, shock, oligouria, polyuria, and convalescent) and can be divided into four severity degrees (mild, moderate, severe, and critical) [5]. These stages are usually classified as the acute phase (including febrile, hypotensive, and oliguric stages) and the convalescent phase (including diuretic and convalescent stages) according to their characteristics of immune responses [6,7]. At the moment, there are no effective therapies available and only limited to supporting treatment [8]. Therefore, studying the immune response against HTNV may be helpful for finding new therapeutic ways.

Although the innate and adaptive immune responses are carefully orchestrated to mediate virus clearance, the T cells mediated anti-viral immune response had core position during HTNV infection. The expressions of CD4 and CD8 molecules on mature CD3^+^T cells are mutually exclusive, defining as helper (CD4^+^T) and cytotoxic (CD8^+^T) T cells, respectively [9]. According to our previous studies, both the CD4^+^T cells and the CD8^+^T cells proliferated in the peripheral blood of HFRS patients after HTNV infection and mounted the protective immune responses [10,11]. Some reports also pointed that CD8^+^T cells promoted viral replication and were involved in the tissue pathogenesis [12]. However, the studies on the CD4^+^ and CD8^+^ double positive T (DP T) cells in HFRS patients are rare. It is noted that DP T cells were progenitor cells in the thymus originated from CD4^−^CD8^−^ double-negative thymocytes [13,14]. It is unexpected to see DP T cells in the peripheral. However, there are still a small proportion of DP T cells existed in the peripheral blood, ranging from 1% to 3% [15]. Under physiological scenarios, DP T cells expanded progressively with age, especially in the individuals over 65 years [16]. However, there was no significant difference between genders [17]. Under pathological states, the number of DP T cells can become elevated in different diseases. It was reported that during human immunodeficiency virus (HIV) [18], human herpesvirus 6 [19], human T cell leukemia virus (HTLV-I) [20], and lymphocytic choriomeningitis virus (LCMV) infections [21], the DP T cells were increased in the peripheral blood. However, the pathophysiological significance of these DP T cells was unclear. Some studies suggested that DP T cells were involved in the disease pathogenesis. In simian immunodeficiency virus (SIV) infection, DP T cells were susceptible to infection and promoted virus entry [22]. During HIV infection, the co-expression of CD4 molecule on CD8^+^T cells promoted DP T cells susceptible to HIV infection and destruction [18]. During Dengue virus infection, frequency of DP T cells was significantly increased. Instead of against virus, they were believed to contributed to the pathogenesis of plasma leakage [23]. In rheumatoid arthritis (RA), DP T cells were recognized as inflammatory cells and contributed to disease pathogenesis [24]. While in other diseases, DP T cells were expected to play a protective role. It was reported that the DP T cells increased, activated and displayed degranulation activity in chronic chagasic patients [25]. In melanoma, DP T cells activated and produced unique cytokine profile. It is believed that DP T cells participate in anti-tumor immune responses in vivo [26]. Eljaafari et al. isolated DP T cells from a skin biopsy of a patient suffering from acute graft-versus-host disease. They defined DP T cells as regulatory T cells with similar gene expression profile of regulatory T cells and high levels of IL-10 expression [27]. In the renal Cell Carcinoma patients, the tumor infiltrating lymphocytes contained an increased proportion of DP T cells. These cells were proved to be dysfunctional tumor-specific T cells and could be set as targets of checkpoint inhibitors [28]. However, in HFRS patients, it still remains unknown about the kinetic changes and potential role of DP T cells during HTNV infection.

In this study, we combined scRNA-seq data with flow cytometry experiments to reveal the phenotypes of DP T cells during HTNV infection. Therefore, this study not only depicts the cellular components and gene expression profiles of DP T cells in the circulating blood of HFRS patients, but also extend the knowledge on the potential roles of DP T cells’ in HTNV infection.

## 2. Materials and Methods

### 2.1. Study Cohort and PBMC Isolation

A total of 66 samples from 59 individuals were enrolled with ages ranging between 17 and 74 years were enrolled in this study. Another 37 healthy donors were collected from physical test. The information of enrolled subjects is summarized in Table 1. The HFRS patients were recruited at the Tangdu Hospital of the Fourth Military Medical University (Xi’an, China) and Xi’an eighth hospital from November 2019–January 2021. Clinical diagnosis of HFRS was confirmed by the detection of HTNV-specific IgM or IgG antibodies. Peripheral blood samples and plasma were collected as previously described.

The medical record number, general clinical information, clinical symptoms and biochemical examination results were recorded in detail. All procedures were in accordance with the ethical standards of the responsible committee on human experimentation (Xijing Hospital, First Affiliated Hospital of Fourth Military Medical University, Xi’an, China). All the data were analyzed anonymously. Each participant provided informed consent.

### 2.2. Single Cell RNA Sequencing

The peripheral blood mononuclear cells (PBMCs) were isolated from the whole blood of HFRS patients and health donors and were performed Single cell RNA sequencing as previously described [29]. The sequencing data can be found at Public Gene Expression Omnibus database (GSE161354). The information of the samples used for sequencing is described in Table 2. We performed data analysis, including the pseudotime analysis by the Novel Bioinformatics Co on the NovelBrain Cloud Analysis Platform in accordance with a previously described protocol [29]. We calculated the marker genes of DP T cells by FindAllMarkers function with Wilcox rank sum test algorithm under following criteria:1. lnFC > 0.25; 2. *p* value < 0.05; 3. min.pct > 0.1.

scTCR-seq data was processed using Cell Ranger (version3.1.0,10xGenomics) against the human VDJ reference provided by10xGenomics. We kept cells with at least one productive TCRα or TCRβ chain for subsequent analysis, such as the distribution of CDR3 length, the usage patterns of VDJ gene segments and combination, and the diversity of TCR repertoire. In each sample, if two or more cells had identical CD8 α-β pairs, these T cells were identified as clonal T cells and shared a unique clone type ID. To integrate TCR results with the gene expression data, the TCR-based analysis was performed only for cells that were identified as T cells. We could identify T cell clonotype shared among clusters or samples.

### 2.3. Flow Cytometry

For surface staining, the 2 × 10^6^ PBMCs from HFRS patients and health donors were stained as previously described. Briefly, cells were washed and suspended in 100 μL flow cytometry staining buffer including respective antibodies. After adding the antibodies, cells were incubated at 4 °C in the dark for 30 min, followed by washing once with staining buffer at 4 °C. Next, cells were resuspended in 100 μL staining buffer and then subjected to flow cytometry.

For intracellular cytokine staining, 4 × 10^6^ PBMCs were stimulated with 20 μg/mL PMA, 2 μM ionomycin and monensin for 4 h at 37 °C. The antibody specific to CD107a was added during PBMCs stimulation. Cells were then washed with staining buffer, followed by surface staining. Next, cells were fixed and permeabilized using an intracellular staining kit (eBioscience, San Diego, CA, USA). The cells were then incubated with antibodies in permeabilization buffer at 4 °C for 30 min. After washing twice, cells were resuspended in 100 μL staining buffer. Intracellular staining of Ki67 was performed as described above but without stimulation. All procedures were performed according to the manufacturer’s instructions. Flow cytometry was conducted on an ACEA Novo Express system (Agilent Bio, Santa Clara, CA, USA) and data was analyzed using the FlowJo software (TreeStar, Woodburn, OR, USA). All the antibodies used in flow cytometry are summarized in Appendix A.

### 2.4. In Vitro Infection

A total of 2 × 10^6^ PBMCs were seeded in 24-well plate and were exposed to HTNV (76–118 strain, kindly provided by the Department of Microbiology, School of Basic Medicine, Fourth Military Medical University) at multiplicity of infection (MOI) of 1 for 2 h, or left unstimulated at 37 °C. Next, cells were washed and then cultured. Seventy-two hours post infection, the cells were collected and stained for flow cytometry analysis. The nucleocapsid protein (NP) of HTNV in DP T cells was detected by flow cytometry intracellular staining following the protocol published previously [12].

### 2.5. Statistical Analysis

All data analyzed were performed using the GraphPad Prism 8 software. All data were presented as a mean value ± standard error of the mean (SEM) based on more than three independent experiments. The Kolmogorov–Smirnov test was carried out to test for normality. Comparisons between or among different groups were performed using Student *t* test or one-way analysis of variance (ANOVA) or two-way ANOVA, respectively. Paried t test was used to analyze paired differences. Correlations were assessed using Pearson analysis. In all tests, values of * *p* <  0.05, ** *p* <  0.01, and *** *p* <  0.001 were considered statistically significant.

## 3. Results

### 3.1. DP T Cells Increased in HFRS Patients

We performed flow cytometry by gating CD3^+^CD4^+^CD8^+^T cells in CD56^−^CD16^−^CD19^−^ PBMCs as DP T cells (Appendix A). As shown in Figure 1A, the representative flow cytometry figures showed that the DP T cells were enriched in HFRS patients. The statistical analysis showed that both the percentage of DP T (Figure 1B) and the number of cells (Figure 1C) increased significantly in HFRS patients. The further analysis showed that the DP T cells were significantly higher in acute phase of HFRS (Figure 1D,E). Although the mild/moderate HFRS patients had higher levels of DP T cells, there was no significant difference between different severity groups (Appendix A). We further investigated the association of DP T cells in HFRS patients with disease severity. The correlation results showed that the percentage of DP T cells had negative correlation with platelet counts (PLT), while the cell counts of DP T cells had positive correlation with blood urea nitrogen (BUN) (Appendix A). Of note, the DP T cells can be separated into CD8^hi^ subset and CD8^lo^ subset, as indicated in Figure 1A. The statistical analysis showed that DP T cells in HFRS patients primarily harbored CD8^hi^ cells (Figure 1F). Conversely, the main population of DP T cells in NC is CD8^lo^ cells (Figure 1F). These results indicated that the frequency of DP T cells increased significantly in HFRS patients, especially in the acute phase of the disease.

### 3.2. The Gene Expression Profile of DP T Cells

To further verify our findings, we performed scRNA-seq on isolated PBMCs. We termed DP T cells by the simultaneously expression of *CD3ε, CD8α* and *CD4* genes (Appendix A). As expected, these DP T cells did not contain NKT cells (Appendix A). The DP T cells can be further classified into 4 clusters (Figure 2A). Compared with normal controls, HFRS patients had more DP T cell counts and had more abundant cell clusters (Figure 2B). We further explored the gene expression profile of each cluster in DP T cells (Figure 2C). Cluster 0 had higher gene expression of *LAG3*, and *KLRC1* genes. It was reported that KLRC1 may contribute to functional CD8^+^ T cell exhaustion [30]. Cluster 2 mainly expressed *TCF7, IL7R* and *KLRB1* genes. It is reported that the combination of IL-7 and IL-7R is important for the early development of T cells in the thymus and their survival in the surrounding areas [31]. Cluster 1 had higher levels of *MKI67*. Cluster 3 mainly expressed genes related to RNA processing and protein production, such as *SLBP* (Stem-Loop Binding Protein), which was an RNA-binding protein involved in the histone pre-mRNA processing [32]. We speculated that cluster 1 was related to T cell activation and proliferation, cluster 0 was associated with T cell exhaustion, while cluster 2 was related to T cell early development and differentiation. The heatmap presenting the KEGG pathway of each cluster was shown in Figure 2D. Cluster 0 mainly activated in immune cells activation pathways and TNF signaling pathway, which plays a role in activation-induced cell death. Cluster 1 and 3 primarily activated in RNA transcription and protein production. These results also proved that cluster 1 and 3 were activated T cells, while cluster 0 was activated and toward exhausted T cells. Cluster 2 only had ribosome activation and may be the steady state T cells.

According to Figure 1, the DP T cells can be divided into CD8^hi^ and CD8^lo^ subsets. We also valued the *CD8α* gene expression in DP T cells based on scRNA-seq data. As we shown in Figure 2E, *CD8α* was lower in cluster 2, but was higher in the other three clusters. Additionally, the bar graph indicated that the DP T cells from normal people were mainly from cluster 2 (Figure 2F). We further performed pseudotime analysis of DP T cells to help understanding the orders of these clusters (Figure 2G,H). In normal controls, it is clear to see that the DP T cells were at the initiation of differentiation, implying these DPT cells were at steady state. After HTNV infection, the DP T cells were at the late stage of differentiation, indicating that these cells differentiated into the functional T cells or exhausted T cells. These results suggested that the cluster 2 was the steady state of DP T cells. HTNV infection led to upregulation of CD8 in the DP T cells, drove DP T cells activation and differentiation to functional T cells and even exhaustion T cells.

The differentiation of mono-positive T cells was tightly regulated by transcriptional factors, particularly ThPOK and Runx3. Runx3 and ThPOK expression is believed to be mutually exclusive [33]. ThPOK suppressed expression of *Runx3* and CD8 lineage genes [34]. Conversely, Runx3 is of critical for CD8^+^ lineage [35]. To understand the development of DP T cells in HTNV infection, we studied the gene expression of ThPOK and Runx3 in HFRS patients based on scRNA-seq. As shown in Appendix A, most DP T cells were Runx3^+^ cells, with only individual cells were ThPOK^+^ cells. Based on the functional phenotypes and the transcriptional factors, we inferred that the DP T cells in the peripheral blood of HFRS patients were from CD8^+^T cell lineage.

### 3.3. DP T Cells Activated and Played Anti-Viral Role in HFRS Patients

To further understand the role of DP T cells played in HTNV infection, we used flow cytometry to analyze the functional phenotypes of DP T cells. The higher percentage and cell counts of Ki67^+^DP T showed that more DP T cells were proliferated in HFRS patients (Figure 3A–C). We also compared the proliferation of DP T cells with that of conventional CD3^+^CD8^+^T and CD3^+^CD4^+^T cells. The results indicated that, DP T cells had similar Ki67 changing trend with conventional CD8^+^T cells (Appendix A). Then, the analysis of cell phenotype showed that these DP T cells in HFRS patients were mainly effector memory T cells (CD45RA^−^CCR7^−^), while in normal controls, DP T cells were mainly naïve T cells (CD45RA^+^CCR7^+^) (Figure 3D,E).

We further analysis the cytokines production of the DP T cells. The flow cytometry results showed that the DP T cells in HFRS patients synthesized significant higher level of granzyme B (GrB) than normal controls. There was no difference in the production of interferon γ (IFN-γ) between them (Figure 4A–C). We also measured the degranulation of DP T cells by staining the surface expression of CD107a. The upregulation of CD107a proved that DP T cells in HFRS patients had better cytotoxic potential than normal controls (Figure 4D,E). After comparing the production of GrB and CD107a between DP T cells and conventional T cells, we noticed that DP T cells exhibited a similar cytokine production profile with CD8^+^T cells (Appendix A). The heatmap based on scRNA-seq data showed the mediators’ production by DP T cell subset in each sample (Figure 4F). The results indicated that the anti-viral mediators, such as granzymes, were the major cytokines produced in DP T cells in HFRS patients.

To gain insight into the anti-viral role of DP T cells, the TCR repertoire of DP T cells were sequenced. Although the frequency of TCRαβ^+^DPT cells decreased in HFRS patients, the majority DP T cells had TCRαβ chains (Appendix A). As shown in Figure 4G,H, compared to normal controls, the HFRS patients had more diversity αβ TCR repertoire distribution. Above results confirmed that DP T cells preformed similar function of CTL cells during HTNV infection.

## 4. Discussion

Our study firstly reported an increased frequency and cell counts of DP T cells in hantavirus diseases. We combined flow cytometry results and scRNA-seq data and showed that DP T cells originated from CD8^lo^ DP T subset. HTNV infection drove these DP T cell proliferation, activation and differentiation to CD8^hi^ DP T subset. These DP T cells in HFRS patients produced higher levels of GrB and CD107a and had more abundant TCR repertoire distribution. Based on these, we speculated that DP T cells in HFRS patients played an anti-viral role.

Previous reports showed that the percentage of DP T cells in PBMCs can increase from 1% up to 20% under stimulations [36,37]. In our study, we gated on the CD3 molecule first. The percentage of DP T cells among CD3^+^T cells was 5% in normal donors, with around 50% CD3^+^T cells containing in PBMCs. After multiplying the DP T% by CD3%, the percentage of DP T in PBMCs of normal controls was around 2.5%. Using this method, the mean percentage of DP T cells in HFRS patients was around 10% in PBMCs. Thus, our data is in accordance with the published data.

Our results suggested that DP T cells in HFRS patients were from CD8^+^T cells lineage. However, the role of co-expression of CD4 antigen on CD8^+^T cells is unclear. It was reported before that CD4 could modulate CTL responses directly by acting as chemotactic receptor, inducer of cellular activation, and adhesion molecule [21]. We speculated that the DP T cells may set as a small CTL pool. When infection occurs, they upregulated their CD8 molecule expression, proliferated quickly to resist the intrusion of pathogens. Our previous study described a subset of GrB^+^CD4^+^T cells displaying adequate cytolytic capacity and promoting HTNV control by gating on CD3^+^CD4^+^ T cells [10]. According to this study, these GrB^+^CD4^+^T population might contain DP T cells and should be categorized as CD8^+^T cells with CD4 molecules expression simultaneously. However, little is known about the mechanisms underlying the maintaining *Cd4* gene expression in DP T cells. Further studies combining single-cell assay for transposase-accessible chromatin sequencing (scATAC-seq) on DP T cells to reveal the epigenetic mechanisms controlling *Cd4* expression in mature CTL cells are needed.

In regard to the source of the peripheral DP T cells, Mizuki et al. studied leukemia patients and suggested that the DP T cells could escape from thymocytes [38]. Flamand et al. found that in HIV infection, TCR stimulation of CD8^+^T cells led to co-expression of CD4 molecule on the cell surface, which caused susceptibility of CD8^+^T cells to HIV infection [39]. They attributed co-expression of CD4 and CD8 on mono-positive T cells to antigenic over-stimulation, which was also applied to our study. To explore the mechanisms underlying the activation of DP T cells, we performed in vitro HTNV infection to see whether DP T cells could be induced in vitro. However, there was no significant difference between unstimulated group and HTNV infection group (Appendix A). The CD8^lo^ and CD8^hi^ subsets of DP T cells were also found to be unchanged (Appendix A). It is possibly because that the HTNV cannot infected DP T cells successfully by detecting nucleocapsid protein (NP) of HTNV expression inside DP T cells (Appendix A) [12]. It is evident that in vitro studies cannot totally mimic the true situation in clinical patients. Misme-Aucouturier et al. reported the kinetic changes of DP T cells post challenge with *Candida albicans*. They found that 6 days after the initial challenge induced significantly increase of DP T cells [40]. Therefore, the infection duration and viral load will influence the final results in vitro. Besides the direct HTNV infection, it is also possible that the DP T cells were activated through bystander pathway. However, the antigen specificity of DP T cells was unknown. Further studies working on the antigen specificity of DP T cells and the activation mechanisms of DP T cells are needed.

Although we proved the antiviral potentials of DP T cells in our study, the roles of DP T cells during HTNV infection were still need further study. Yu et al. identified the DP T cells in Dengue virus infection as a potential marker of plasma leakage progression to shock and may contribute to the pathogenesis of disease [23]. We also studied the correlation between dynamic changes of DP T cells and clinical indicators of HFRS patients. The results indicated that the amount of DP T cells had the negative correlation with PLT and positive correlation with BUN (Appendix A). According to this, we speculated that the DP T cells in HFRS might be involved in the pathogenesis of hemorrhage and renal failure by producing GrB and CD107a to disrupt the endothelial cell barrier. The further studies including in vitro co-culture assays are needed to understand the roles of DP T cells played during HTNV infection.

However, this study had some limitations, including the lack of suitable animal disease model and the limitation of sample collection. Consequently, the origin of DP T cells was not fully elucidated. In addition, the antigen specific of DP T cells are unclear. Therefore, future studies on how to specifically induce the anti-viral DP T cells in order to control HTNV infection are needed.

## 5. Conclusions

In summary, based on the kinetics changes, gene expression profile and cytokine productions, the DP T cells participated in the adaptive immune response against HTNV. By unraveling mechanisms underlying the cellular orchestration of these cells throughout HTNV infection, it may be possible to manipulate immune responses and enhance effector function.

## Figures and Tables

**Figure 1 viruses-14-02243-f001:**
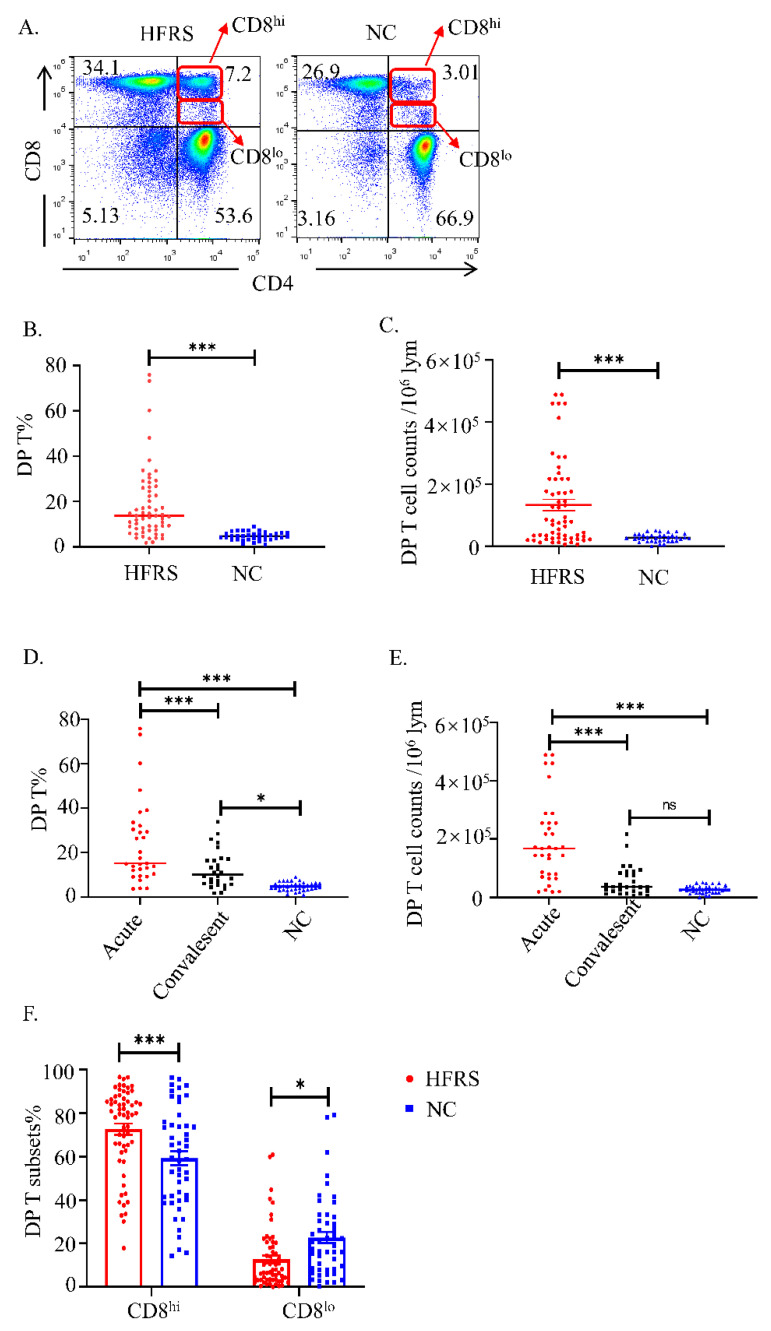
The kinetic changes of DP T cells in HFRS patients. (**A**). The representative flow cytometric plots of DP T cells in both HFRS patients and normal controls (NC). The gradually changing of the dots ‘colors indicated the cell density. The red color indicated the high cell density, while the blue color indicated the low cell density. Summary data of the comparison of the (**B**) percentage and (**C**) cell counts of DP T cells in HFRS patients and NC. Summary data of the comparison of the (**D**) percentage and (**E**) cell counts of DP T cells in different phases (acute phase and convalescent phase) of HFRS patients and NC. (**F**) The comparison of CD8^hi^ and CD8^lo^ subsets of DP T cells in HFRS patients and NC (For HFRS, *n* = 59; For NC, *n* = 36). * *p* < 0.05; *** *p* < 0.001.

**Figure 2 viruses-14-02243-f002:**
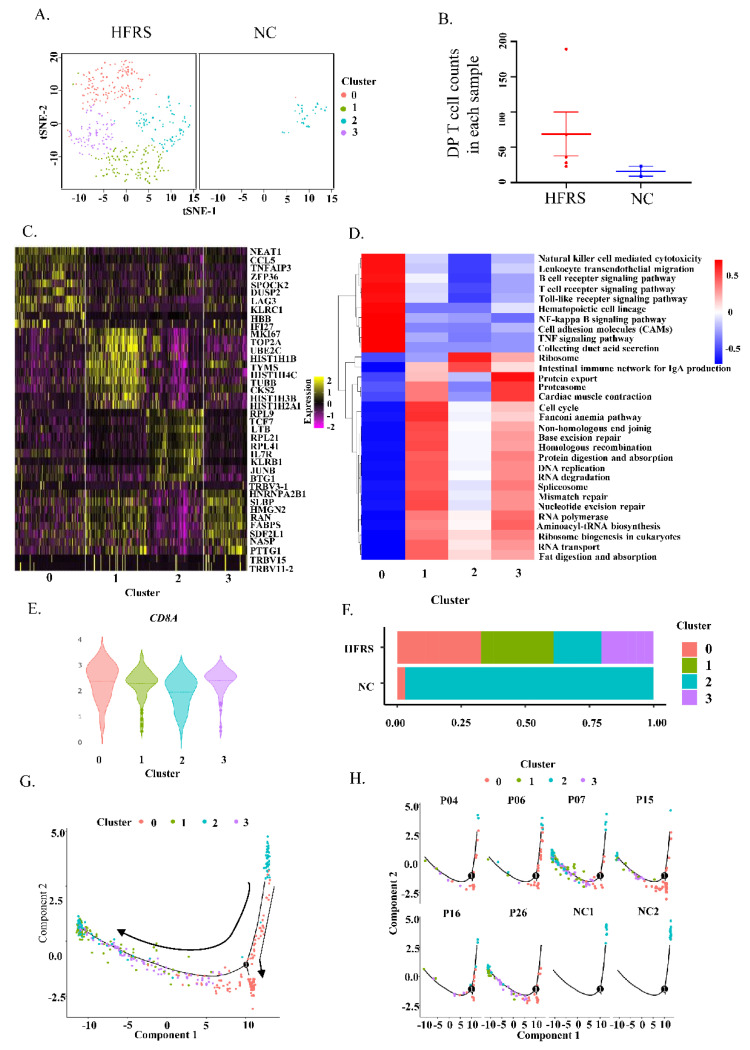
Identified DP T cells in HFRS patients by scRNA-seq. (**A**). t-SNE plot showing DP T cells from both HFRS and NC. (**B**) Dot plot showing the comparison of DP T cell counts in each sample of HFRS and NC. (**C**) The heatmap presenting gene expression profile of each cluster in DP T cells. (**D**) The KEGG heatmap displaying the pathway activation in each cluster of DP T cells. (**E**) The violin graph showing the comparison of *CD8α* gene expression in each cluster. (**F**) The bar graph presenting the cluster distribution in both HFRS and NC. (**G**) The pseudotime trajectory analysis of DP T cells. (**H**) The pseudotime trajectory analysis of DP T cells in each sample.

**Figure 3 viruses-14-02243-f003:**
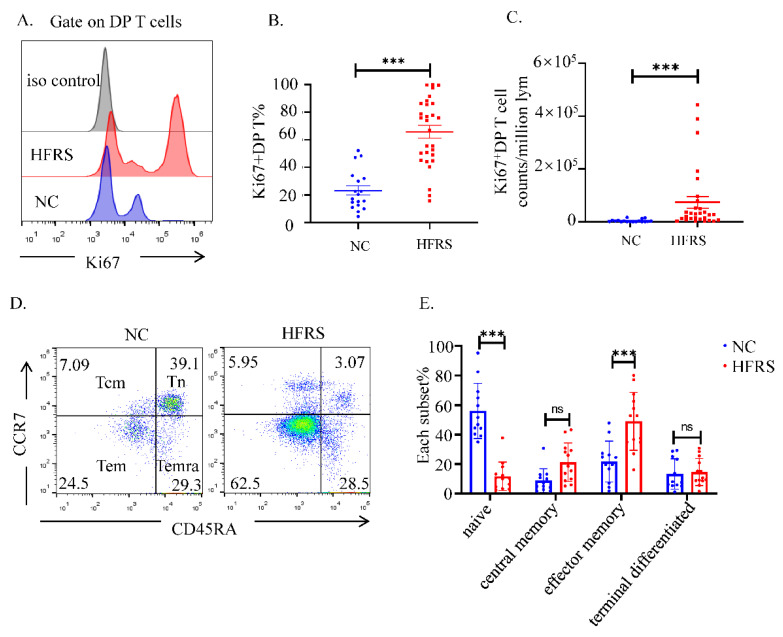
The activation status of DP T cells. (**A**). The expression of Ki67 in DP T cells was evaluated by flow cytometry. The representative histogram figures of Ki67 expression in DP T cells from both HFRS patients and NC, setting iso type staining as controls. (**B**) Comparison the percentage of Ki67^+^DP T cells and the (**C**) MFI of Ki67 in DP T cells from both HFRS patients and NC (For HFRS, *n* = 29, For NC, *n* = 18). (**D**) T cell subsets are defined as naïve (Tn) (CD45RA+, CCR7+), central memory (Tcm) (CD45RA−, CCR7+), effector memory (Tem) (CD45RA−, CCR7−), and terminal effectors (Temra) (CD45RA+, CCR7−). The representative dot blot figures showed the activation status of T cell subsets in DP T cells. (**E**) The statistical analysis showed the comparison of each T cell subset in both HFRS patients and NC (For HFRS, *n* = 13. For NC, *n* = 12). *** *p* < 0.001; “ns” means no significance.

**Figure 4 viruses-14-02243-f004:**
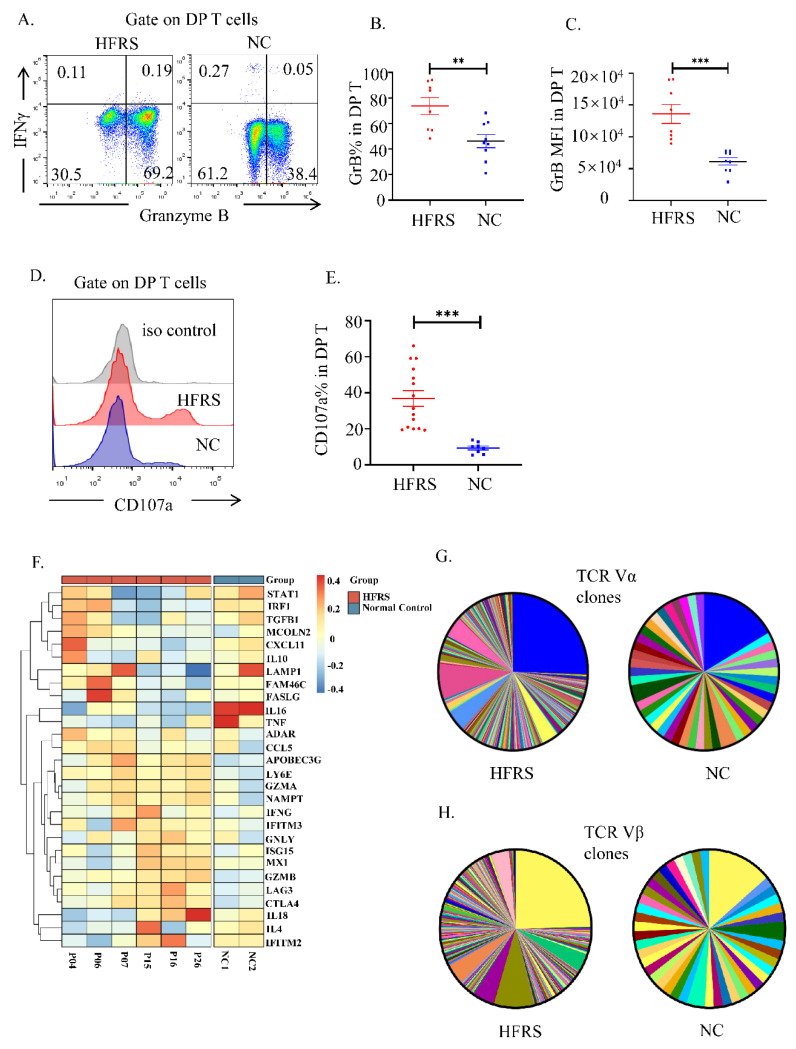
The cytotoxic characteristics of DP T cells. (**A**). The representative flow cytometric plots showed the IFN-γ and granzyme B production of DP T cells. The gradually changing of the dots ‘colors indicated the cell density. The orange color indicated the high cell density, while the blue color indicated the low cell density. (**B**,**C**) The statistical analysis showed the comparison of (**B**) GrB% and (**C**) GrB MFI in both HFRS patients and NC (*n* = 9). (**D**) The representative histogram showed the expression of CD107a of DP T cells. (**E**) The comparison of CD107a% in both HFRS patients and NC (For HFRS, *n* = 15; For NC, *n* = 8). (**F**) The heatmap summarized the mediators’ production in DP T cells in each sample of scRNA-seq. (**G**,**H**) The pie chart presented the comparison of TCR repertoire analysis of TCR Vα (**G**) and TCR Vβ (**H**) in HFRS patients and NC. Each color represented a unique sequence of Vα or Vβ. ** *p* < 0.01; *** *p* < 0.001.

**Table 1 viruses-14-02243-t001:** Characteristics of enrolled subjects.

	Mild/Moderate	Severe/Critical	NC
**Demographic characteristics**			
number	21	38	37
Age (years)	43 (17–74)	45 (20–70)	36 (21–55)
Male (%)	82.6%	89.1%	70.0%
**Sample number**	23	43	
Acute phase (febrile/hypotensive/oliguric)	11	24	_
Convalescent phase (diuretic/convalescent)	12	19	_

NC: normal controls. Values represent medians with the corresponding interquartile range.

**Table 2 viruses-14-02243-t002:** Information of subjects enrolled in scRNA-seq.

Sample No.	Age	Sex	Disease Phase	Disease Severity	DP T Cell Counts/Total Sequencing Cell Counts
P04	52	male	oligouria	critical	28/4556
P06	34	male	oligouria	moderate	36/3746
P07	49	male	fever	moderate	189/4218
P15	29	male	fever, shock, and oligouria	critical	68/4551
P16	37	male	fever	moderate	23/4129
P26	26	male	fever, and shock	critical	66/3763
NC1	29	male	/	/	9/4253
NC2	33	female	/	/	23/3660

P: patient of HFRS. NC: normal control. DP T cell counts: The DP T cell counts in each sample for Single cell RNA sequencing. total sequencing cell counts: The total cell counts in each sample for Single cell RNA sequencing.

## Data Availability

The authors declare that all data supporting the findings of this study are available within this article and its Appendix A, or from the corresponding author upon reasonable request. Single-cell RNAseq gene expression data have been deposited in the Gene Expression Omnibus database (GSE161354).

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
