# Peer review of "Increased CD4+CD8+ Double Positive T Cells during Hantaan Virus Infection"

_viruses, 2022, doi:10.3390/v14102243_

Round 1

Author Response

In this article, the authors will explore the rare population of double-negative T lymphocytes (DPTs)

in patients infected with Hantaan virus by comparing individuals with different degrees of disease

with uninfected individuals. The explored markers by flow cytometry and ARN sequencing. It is

shown an increased on DPTs in infected individuals and some gene expression can be divided the

cells in clusters. This is a well conducted and written manuscript, however, some changes and

clarifications should be made.

We thank the reviewer to give us the opportunity to improve our manuscript. The point-by-point reply is as follows. We have also learned a lot from this process. We tried our best to accomplish the experiments. However, because it is not in the epidemic season, it is difficult to collect fresh PBMC samples from HFRS patients. So all the assays were performed using our very limited stored PBMCs. We have replied the reviewers’ comments point by point. We hope it could meet the criteria for publication this time.

Introduction

Point 1. In the introduction other elements should be mention such as studies on DPT reference

values and changes in populations according to age or gender.

Response 1. We thank the reviewer for these very constructive suggestions. We have added more references related to the DP T cells changes in the introduction section.

Methodology

Point 2. This reviewer did not have access to the supplementary table with the markers used for

flow cytometry, it is recommended to mention at least the list CD markers in methodology.

Response 2. We thank the reviewer for the helpful suggestions. We are sorry for our negligence. We have added a supplementary table 1 listed all the markers used in the flow cytometry.

Point 3.  In statistics, it is assumed that a normality test was carried out since parametric statistics

are applied.

Response 3. We thank the reviewer for the constructive suggestions. We have added the normality test in the statistical analysis.

Results.

Point 4. In Table 2, The authors should clarify the meaning of cell count and mentioned as footnote.

Response 4. Thanks for the comments. Because the frequency of a cell subset may keep unchanged when the whole lymphocytes were expanded during disease. So the parameter of total cell counts was used to reflect the kinetic changes of absolute cells in the total sequencing cells from another angel. We have added the meaning of cell counts as footnote in the Table 2.

Point 5. Figure 1. Please pay attention to dot plot figure of DPT cells in non-infected donor who has

a very low number of CD8+ T cells, at least in percentage compared to CD4+ T cells.

Response 5. We are glad the reviewer to point out this critical issue. We are sorry for our negligence. In normal people, the ratio of CD4+T and CD8+T should be around 2:1. There was individual difference among each sample. We changed Figure 1A to a more representative dot plot of the normal people.

Point 6. Are the percentages of both populations of DPT cells shown in Figure 1F from normal donors

in accordance with published data?

Response 6. We thank the reviewer to give this creative suggestion. In our results (Fig 1B), the percentage of DPT cells among CD3+T cells is 4.95±0.33 in normal donors, with around 50% CD3+T cells containing in PBMCs. After multiplying the DPT% by CD3%, the percentage of DPT in PBMCs is around 2.5%. We compared our results with previously published data.  The reported percentage of DPT cells in PBMCs is about 1%-3% (Blue et al., J Immunol, 1985 Apr, 134 (4).; Yu et al., Viruses, 2022, 14, 90., Giraldo et al., PLoS Negl Trop Dis. 2011 Aug;5(8):e1294). According to the above, our data is in accordance with the published data. We have added the explanation of our data in the discussion section.

Point 7. Do the authors suggested three (3) clusters of DPTs, but then mentioned from zero (0) to

four (4). It needs to be clarified if they are 3 o 4 clusters.

Response 7. We thank the reviewer to point out our mistakes. We have made the corresponding corrections.

Point 8. It is very confusing the color code for CD8α expression in figures 2E to H, it does not allow

an easy followed up for the reader. Do the authors think is necessary figure 2G and 2H to

make a point?

Response 8. We are sorry for the confusion we made. We have unified the colors in each cluster. The pseudotime analysis in Figure 2G-H is helpful in understanding the gene changes during the DPT cells’ differentiation. And it is clearly to see in Figure 2H that in normal controls, the DPT cells were at the initiation of differentiation. Conversely, the DPT cells in HFRS patients were at the late stage of differentiation. We have made our point more clearly in our manuscript.

Point 9.  Do the authors used any stimulation to measure IFN γ by flow cytometry in DPTs cells?

Response 9. We thank the reviewer for asking this key question. As we stated in the methods section, we performed intracellular cytokine staining by stimulating the 4×106 PBMCs with 20 μg/ml PMA, 2 μM ionomycin and monensin (1:1000) for 4 h at 37℃ at first. Then the staining process was conducted as following.

Point 10. It will be informative if the mentioned the percentage of most common TCRβ usage.

Response 10. Thank you for your good comment. There is no commercial antibody specific to human TCRβ chain. We tried to use antibody specific to mouse TCRβ chain, which has been reported to have cross reaction with human TCRβ chain (Clone No. H57-597). However, the specificity and the affinity of this mouse antibody is poor in reaction to human samples. So, we stained the expression of TCRα/β chain in DPT cells in PBMC samples (the antibody information has been updated to supplementary table 1). The results showed that over 90% DP T cells are αβT cells in normal controls. However, compared to normal donors, the frequency of TCRαβ+ DP T cells in HFRS patients decreased. We have updated the results in Supplementary Figure 4.

Point 11. In general, figure legend should be more informative.

Response 11. We thank the reviewer for the constructive comment. We have added more details in each figure legend.

2

Supplementary figures:

Point 12. In Figure S1 D and E, it will be nice to have the regression line. Please mentioned if they do

plot belongs to a patient of normal donor: DPT% 3.18.

Response 12. Thank the reviewer to point out this important issue. We have added the regression line in Sup Fig 1 D-E. These plots all belong to HFRS patients. We have made it more clearly in the manuscript.

Point 13. In Figure S2, please indicate in the figure legends what numbers 0,1, 2 and3 are indicating.

Response 13. We are sorry for our negligence. These numbers indicated each cluster in DPT cells. We have added details in the figure legends.

Discussion

Point 14. Are the CD8+low DPT cells increased expression during viral infection?

Response 14. We thank the reviewer to ask this question. There is no significant change of DPT subset during in vitro infection. We added the comparison of CD8hi and CD8lo DPT cell subsets between unstimulated group and HTNV infection group in Sup Fig 4B.

Point 15. They are studies published on DPT values which are not mentioned, as well as discussing in the

context or other infectious models

Response 15. We are sorry for our negligence. We have enriched the content in the introduction and discussion sections with the following related references discussing the DPT values in the other infectious models.

Point 16. Could the DPTs cells in Hantaan virus infection be protective or part of the pathogenesis?

Response 16. We thank the reviewer to ask this question. The role of DPT cells in HTNV infection is discussed in the discussion section. Based on our results, we tend to think that DPT cells played an anti-viral role. However, the correlation between DPT cells and clinical parameters indicated that DPT cells might also contribute to the disease pathogenesis. The scRNA-seq data for effector cytokines and anti-viral mediators were added in Figure 4 in the latest version. Based on these, we thought that DP T may contribute to the pathogenesis by expression Granzymes. These mediators are thought to disrupt the endothelial cell barrier by cytotoxicity. We have added this point in our discussion. The further in vitro studies coculturing DPT cells with HTNV target cells are needed to verify the anti-viral role of DPT cells.

Reviewer 2 Report

Broad Comments:

This original manuscript takes an approach in the study of DP T cells, the authors characterized DP T cells from HFRS patients based on flow cytometry data combined with scRNA-seq data. They showed that HTNV infection cause upregulation of DP T cells in the peripheral blood and they correlated this with disease severity. Also, they identified the accumulation of DP T cells in the peripheral blood of HFRS patients and suggested that these DP T cells belong to CD8+T cells lineage. This is the valuable addition to the field of science and immunological medicine. In general, this manuscript is easy to read and follow. The figures and the text throw together. The manuscript will benefit from attending to few specific comments bellow.

Major comment

The authors should consider revising the English throughout the manuscript. The manuscript contains grammatical errors and some of them I have highlighted them bellow.

Specific Comments:

1.    In the introduction typo “Sever”, it should read severe

2.    Materials and methods-the authors should consider revising the first sentence, it is not clear whether the author enrolled the samples or enrolled people.

3.    Also, in the same first sentence it is not clear if 46 samples were obtained from 40 people, if so could the authors state how they got the overlap of 6 samples  

4.    The sentence “was summarized in table 1” should read is summarized in Table 1

5.    The sentence “Clinical diagnosis of HFRS was confirmedly by the detection” should read Clinical diagnosis of HFRS was confirmed by the detection

6.    Invitro section- the authors would like to revise the sentence starting with the number.

7.    Discussion: The authors should consider revising this sentence “HTNV infection drove these DP T cells proliferated, activated and differentiated to CD8hi DP T subset” to “HTNV infection drove these DP T cells proliferation, activation and differentiation to CD8hi DP T subset”  

8.    “These might because that the” this sentence is not clear.

Author Response

2.

The authors should consider revising the English throughout the manuscript. The manuscript contains grammatical errors and some of them I have highlighted them bellow.

We thank the reviewer to give us these many constructive suggestions to improve our manuscript. We have revised the language and replied to the comments point by point. We hope it could meet the criteria for publication this time.

Specific Comments:

  1. In the introduction typo “Sever”, it should read severe

We appreciate very much for pointing out this important issue. We have revised the “Sever” to “Severe” in red.

  1. Materials and methods-the authors should consider revising the first sentence, it is not clear whether the author enrolled the samples or enrolled people.

We appreciate very much for this suggestion. We enrolled samples from different HFRS patients in our study. We have stated it more clearly in the revised version of manuscript.

  1. Also, in the same first sentence it is not clear if 46 samples were obtained from 40 people, if so could the authors state how they got the overlap of 6 samples

We thank the reviewer to ask this important issue. It is because we collected more than one sample from one patient. For example, we collected 66 plasma samples from 59 HFRS patients. 7 of 59 patients collected 2 samples from different phases. We added the illustration for acute phase and convalescent phase and time points for sample collection in “2.1. Study cohort and PBMC isolation” and in Table 1 as well as its reference (6,7) in the revised version of manuscript.

  1. The sentence “was summarized in table 1” should read is summarized in Table 1

Thanks for pointing out this mistake. We have revised the “was” to “is” in the revised manuscript in red.

  1. The sentence “Clinical diagnosis of HFRS was confirmedly by the detection” should read Clinical diagnosis of HFRS was confirmed by the detection

Thank you for your comment. We are sorry for our mistake. We have revised the typo word.

  1. Invitro section- the authors would like to revise the sentence starting with the number.

We appreciate very much for your careful review. We have revised the sentence in the revised version of the manuscript.

  1. Discussion: The authors should consider revising this sentence “HTNV infection drove these DP T cells proliferated, activated and differentiated to CD8hi DP T subset” to “HTNV infection drove these DP T cells proliferation, activation and differentiation to CD8hi DP T subset”

Thanks for pointing out this mistake. We have revised the sentence “HTNV infection drove these DP T cells proliferated, activated and differentiated to CD8hi DP T subset” to “HTNV infection drove these DP T cells proliferation, activation and differentiation to CD8hi DP T subset”.

  1. “These might because that the” this sentence is not clear.

Thank you for your comments. We are sorry for the mistake. We have revised it to “It is possibly because”.

Reviewer 3 Report

1)     The authors describe the increase in double positive CD4+CD8+ T cells during acute human Hantaan virus infection.  This phenomenon would be consistent with the increased prevalence of double positive CD4+CD8+ T cells in the course of other human diseases.  However, the flow cytometry data provided by the authors is weak.  A list of antibodies, including clone numbers, needs to be included in the methods section.  To their credit, the authors use a restrictive lymphocyte gate (perhaps too restrictive as it may underestimate the size of the activated lymphocyte population) and then gate on CD3 positive events.  Increased lymphocyte cell death has been a published phenomenon during human hantavirus infection.  Cell viability is not assessed and can be a major cause of nonspecific binding of antibodies leading to a false double positive phenotype.  In addition to a standard live/dead dye which will detect late apoptosis, annexin V should be used to detect and exclude early apoptotic cells.  It is also crucial that the authors exclude B cells (CD19) and monocytes (CD14) in their gating strategy.  Human monocytes which can express CD4 can form doublets with human T cells leading to the appearance of CD8+CD4+ T cells that will fall in a CD3+ gate.  The use of a CC19 and CD14 dump channel antibodies along with a singlet gate (e.g. FSC-A vs FSC-H) will exclude the possibility of doublets.  Finally, the expression of CD4 and CD8, as well as the gene expression data provided by the authors in Figure 2 are consistent with both human T cells and NKT cells.  The authors need  to include a marker of NKT cells such as CD161 or CD56 in their analysis to preclude the possibility that they have identified an increase in NKT cells during Hantaan virus infection, not double positive CD4+CD8+ T cells.  An increase in NKT cells during Hantaan virus infection would also be interesting but would require a more extensive rewrite of the manuscript.  As it is, as revised gating scheme will likely alter the percentages of double positive cells if the authors have indeed identified double positive T cells.  A good example of an appropriate gating scheme is shown in reference 18 provided by the authors.  If this gating scheme was used by the authors, it needs to be mentioned in the methods section and fully diagramed either in Figure 1 or in Supplementary Figure 1.      

2)    The authors provide a beautiful set of RNA-seq data.  Again, as mentioned earlier, the genes highlighted are shared between conventional T cells and NKT cells.  It must be demonstrated that the cells the authors are analyzing are indeed double positive T cells and not NKT cells.  Additionally, it is puzzling why the authors did not include an analysis of cytokine, chemokine and other anti-viral mediators in their analysis of this cell population.  Vascular permeability is the hallmark of hantavirus infection and it would be extremely important to understand if this expanded double positive T cell or NKT cell population is expressing genes for cytokines and anti-viral mediators that might influence vascular permeability or endothelial dysfunction (ie IL-1, IL-6,  TNFa, IL-17, granulysin, etc) or chemokines that might orchestrate inflammation.

3)    The authors need to be careful not to overextend their interpretation of their Ki-67 results.  Ki-67 is an indication that a cell has proliferated at least one time within approximately 30 days.  The authors claim that the DP T cell population has an enhanced proliferation capacity than conventional T cells.  The use of Ki-67 to make this statement is not accurate.  It is true that a higher percentage of DP cells have proliferated.  This would be expected as the size of the DP population, which is already significantly smaller than the size of the conventional CD4 or CD8 signal positive T cell populations, increases during infection.  In order to show an enhanced proliferation capacity, the authors would have to compare the number of cell divisions that the DP population undergoes during infection compared to the number of cell divisions that the conventional T cell population undergoes during the same time period using something like CFSE or a Cell Trace dye.  This isn’t possible in a human study where the cells are analyzed ex vivo. 

4)    The authors directly compare DP cells to conventional single positive CD8 T cells in supplementary figure 3.  However, given the disparity in the sizes of the DP T cell and single positive CD8 population, it is not obvious what percentage of conventional CD8 T cells are expressing Ki-67 during acute infection compared to normal healthy controls or convalescent HFRS patients.  A better way of controlling for the size differences of the conventional and DP populations would be to compare the percent of TEM cells (CCR7lo/CD45RAlo) cells, essentially effector cells, in each population expressing Ki-67 in HFRS patient vs Normal Control.  It’s likely that the percent of Ki-67 cells in each population of cells would be much more similar.  This would At minimum, the authors should 1) extend their data set in Figure 3 to include data showing the percent of Ki-67 positive cells for both conventional CD4 and CD8 single positive T cells in HFRS patients compared to Normal Controls, 2) compare the total number of Ki-67 positive cells for both conventional CD4 and CD8 single positive T cells, as well as DP T cells, in HFRS patients compared to Normal Controls.  Total number of Ki-67 cells in each population will likely demonstrate a sizeable increase in conventional CD4 and CD8 single positive T cells, as well as DP T cells, further demonstrating that all three subsets proliferate well in response to HTNV infection.       

5)    In figure 4 the authors demonstrate the function of the DP cell population.  The authors demonstrate that the DP cell population in HFRS patients has a higher percentage of GranzymeB expressing cells than the same cells in normal controls.  1) It should be clarified in the text that this is not a measure of GranzymeB release but is a measure of GranzymeB synthesis upon restimulation.  The cells were treated with monensin which should prevent GranzymeB release.  An ELISA would be necessary to measure released GranzymeB.  If the authors wish to measure degranulation, a degranulation assay examining cell surface expression of CD107a in the absence of monensin would be appropriate. 

6)    The DP cells in figure 4 produce a surprisingly small amount of IFNg.  Have the authors looked at other cytokines potentially produced by these DP cells.   Additionally, the authors should demonstrate how the GranzymeB/IFNg phenotype of the DP population compares to the conventional CD8 and CD4 single positive T cell populations. Figure 4 would also be an appropriate place to include RNS-seq data for effector cytokines and anti-viral mediators.   

7)    Do the authors know if the DP population present in HFRS patients is specific for Hantaan antigens?  Alternatively, is this a population of bystander activated cells?  The authors should discuss this possibility.  It is perhaps beyond the scope of this study but a 24hr restimulation of PBMCs from HFRS patients with recombinant Hantaan virus NP or G1 proteins, followed by flow cytometry analysis of activation induced markers such as OX40, PD-L1, CD25 on T cells, or cytokine bead array analysis of culture supernatants might reveal antigen specificity.  Something similar was done by Van Epps, Schmaljohn and Ennis in 1999. Van Epps HL, Schmaljohn CS, Ennis FA. Human memory cytotoxic T-lymphocyte (CTL) responses to Hantaan virus infection: identification of virus-specific and cross-reactive CD8(+) CTL epitopes on nucleocapsid protein. J Virol. 1999 Jul;73(7):5301-8. doi: 10.1128/JVI.73.7.5301-5308.1999. PMID: 10364276; PMCID: PMC112585.

8)    In the discussion, the authors mention that in vitro HTNV infection did not alter the percentage of DP cells.  However, the authors don’t demonstrate that the cells in culture were even infected with virus.  Is there any evidence that any of the cells in culture were infected with HTNV?  As is, Supplementary Figure 4 is an uninterpretable negative result.  Reference 12 provided by the authors has a nice example of demonstrating HTNV infection using a NTHV-NP antibody and is presumably why this experiment was performed.  This paper should be referenced in this portion of the discussion as it is very relevant.   If cells, particularly monocytes of B cells are infected during this assay, is there evidence that the DP or conventional single positive T cell populations respond to the infected monocytes or B cells?  Again, this would demonstrate antigen specificity.    

9)    In the abstract, the authors mention that, “HTNV infection caused the upregulation of DP T cells in the peripheral blood, which were correlated with disease severity”.  However, in the text for Figure 1, the authors state that, “although the mild/moderate HFRS patients had higher levels of DP T cells than severe/critical HFRS patients, there was no significant difference between different severity groups”.  These two statements do not agree and need to be addressed.

10)   The authors suggest that DP T cells in HFRS might be involved in the pathogenesis of hemorrhage and renal failure.  Please discuss how these cells might contribute to pathogenesis.  Again, including  RNS-seq data for effector cytokines and anti-viral mediators  for the DP population would be incredibly valuable and should be included.

11)   Please discuss what, if anything, is known about the presence of DP T Cells in hantavirus disease caused by other hantaviruses (ie Puumala, Sin Nombre, Andes).

Author Response

Reviwer3

We thank the reviewer to give us many professional suggestions to improve this manuscript. We have also learned a lot from this process. We tried our best to accomplish the experiments. However, because it is not in the epidemic season, it is difficult to collect fresh PBMC samples from HFRS patients. So all the assays were performed using our very limited stored PBMCs. We have replied the reviewers’ comments point by point. We hope it could meet the criteria for publication this time.

1)     The authors describe the increase in double positive CD4+CD8+ T cells during acute human Hantaan virus infection.  This phenomenon would be consistent with the increased prevalence of double positive CD4+CD8+ T cells in the course of other human diseases.  However, the flow cytometry data provided by the authors is weak.  A list of antibodies, including clone numbers, needs to be included in the methods section.  To their credit, the authors use a restrictive lymphocyte gate (perhaps too restrictive as it may underestimate the size of the activated lymphocyte population) and then gate on CD3 positive events.  Increased lymphocyte cell death has been a published phenomenon during human hantavirus infection.  Cell viability is not assessed and can be a major cause of nonspecific binding of antibodies leading to a false double positive phenotype.  In addition to a standard live/dead dye which will detect late apoptosis, annexin V should be used to detect and exclude early apoptotic cells.  It is also crucial that the authors exclude B cells (CD19) and monocytes (CD14) in their gating strategy.  Human monocytes which can express CD4 can form doublets with human T cells leading to the appearance of CD8+CD4+ T cells that will fall in a CD3+ gate.  The use of a CC19 and CD14 dump channel antibodies along with a singlet gate (e.g. FSC-A vs FSC-H) will exclude the possibility of doublets.  Finally, the expression of CD4 and CD8, as well as the gene expression data provided by the authors in Figure 2 are consistent with both human T cells and NKT cells.  The authors need  to include a marker of NKT cells such as CD161 or CD56 in their analysis to preclude the possibility that they have identified an increase in NKT cells during Hantaan virus infection, not double positive CD4+CD8+ T cells.  An increase in NKT cells during Hantaan virus infection would also be interesting but would require a more extensive rewrite of the manuscript.  As it is, as revised gating scheme will likely alter the percentages of double positive cells if the authors have indeed identified double positive T cells.  A good example of an appropriate gating scheme is shown in reference 18 provided by the authors.  If this gating scheme was used by the authors, it needs to be mentioned in the methods section and fully diagramed either in Figure 1 or in Supplementary Figure 1.      

We thank the reviewer for the critical comments. We have followed the suggestions and revised the gating scheme according to reference 18 and updated our data in Supplementary Figure 1 and Figure 1. After checking the previous results, we noticed that we did not gate the monocytes. The co-expression of CD3, CD4 and CD8 was very rare on the B cell subset. Fortunately, we stained CD161 molecule previously. So we deleted the NKT cells from DP T cells as the reviewer suggested. The manuscript about NKT cells in HFRS patients is prepared at present by our group. Our results indicated that the frequency and cell numbers of NKT cells decreased significantly during HTNV infection. Based on these, our previous conclusion still holds.

2)    The authors provide a beautiful set of RNA-seq data.  Again, as mentioned earlier, the genes highlighted are shared between conventional T cells and NKT cells.  It must be demonstrated that the cells the authors are analyzing are indeed double positive T cells and not NKT cells.  Additionally, it is puzzling why the authors did not include an analysis of cytokine, chemokine and other anti-viral mediators in their analysis of this cell population.  Vascular permeability is the hallmark of hantavirus infection and it would be extremely important to understand if this expanded double positive T cell or NKT cell population is expressing genes for cytokines and anti-viral mediators that might influence vascular permeability or endothelial dysfunction (ie IL-1, IL-6,  TNFa, IL-17, granulysin, etc) or chemokines that might orchestrate inflammation.

We thank the reviewer for the very good questions. First, we have highlighted the DP T cells and NKT cells in the whole T/NK/NKT cells and updated the results in Sup Fig 2B. The DP T cells and NKT cells are not in the same cluster. They are not even close. Second, we added the analysis of cytokines in DP T cells. The IL-1 and IL-6 were not expressed in DP T cells.  As shown in updated heatmap in Fig 4, the DP T cells mainly produced anti-viral mediators.

3)    The authors need to be careful not to overextend their interpretation of their Ki-67 results.  Ki-67 is an indication that a cell has proliferated at least one time within approximately 30 days.  The authors claim that the DP T cell population has an enhanced proliferation capacity than conventional T cells.  The use of Ki-67 to make this statement is not accurate.  It is true that a higher percentage of DP cells have proliferated.  This would be expected as the size of the DP population, which is already significantly smaller than the size of the conventional CD4 or CD8 signal positive T cell populations, increases during infection.  In order to show an enhanced proliferation capacity, the authors would have to compare the number of cell divisions that the DP population undergoes during infection compared to the number of cell divisions that the conventional T cell population undergoes during the same time period using something like CFSE or a Cell Trace dye.  This isn’t possible in a human study where the cells are analyzed ex vivo. 

We appreciated very much for your professional comments. We have revised our statements. As you said, the cell division in the same time period using cell trace dye cannot be performed in the current study. We will keep it in mind and test proliferation capacity of DP T cells in our future study.  

4)    The authors directly compare DP cells to conventional single positive CD8 T cells in supplementary figure 3.  However, given the disparity in the sizes of the DP T cell and single positive CD8 population, it is not obvious what percentage of conventional CD8 T cells are expressing Ki-67 during acute infection compared to normal healthy controls or convalescent HFRS patients.  A better way of controlling for the size differences of the conventional and DP populations would be to compare the percent of TEM cells (CCR7lo/CD45RAlo) cells, essentially effector cells, in each population expressing Ki-67 in HFRS patient vs Normal Control.  It’s likely that the percent of Ki-67 cells in each population of cells would be much more similar.  This would At minimum, the authors should 1) extend their data set in Figure 3 to include data showing the percent of Ki-67 positive cells for both conventional CD4 and CD8 single positive T cells in HFRS patients compared to Normal Controls, 2) compare the total number of Ki-67 positive cells for both conventional CD4 and CD8 single positive T cells, as well as DP T cells, in HFRS patients compared to Normal Controls.  Total number of Ki-67 cells in each population will likely demonstrate a sizeable increase in conventional CD4 and CD8 single positive T cells, as well as DP T cells, further demonstrating that all three subsets proliferate well in response to HTNV infection.       

Many thanks for your suggestions. We are sorry for the confusion we made. We rechecked our data and compared the percentage and total number of Ki67 positive cells for both conventional single positive T cells and DP T cells in HFRS patients and normal controls. We have updated these results in Fig 3 and Sup Fig 3. As you said, the percentage of Ki67 cells in each population of cells increased in HFRS. The change of Ki67% in DP T cells and CD8+T cells was much more similar.

5)    In figure 4 the authors demonstrate the function of the DP cell population.  The authors demonstrate that the DP cell population in HFRS patients has a higher percentage of GranzymeB expressing cells than the same cells in normal controls.  1) It should be clarified in the text that this is not a measure of GranzymeB release but is a measure of GranzymeB synthesis upon restimulation.  The cells were treated with monensin which should prevent GranzymeB release.  An ELISA would be necessary to measure released GranzymeB.  If the authors wish to measure degranulation, a degranulation assay examining cell surface expression of CD107a in the absence of monensin would be appropriate. 

We are grateful for your suggestions. We have revised our statement in the manuscript. It’s hard to use ELISA to measure the released GrB from DP T cells from the HFRS patients. Culturing and stimulating the sorted DP T cells in vitro may alter the status of DP T cells. We have also added the degranulation assay by examining cell surface expression of CD107a. The updated results in Fig 4D-E indicated the increasing level of CD107a.

6)    The DP cells in figure 4 produce a surprisingly small amount of IFNg.  Have the authors looked at other cytokines potentially produced by these DP cells.   Additionally, the authors should demonstrate how the GranzymeB/IFNg phenotype of the DP population compares to the conventional CD8 and CD4 single positive T cell populations. Figure 4 would also be an appropriate place to include RNS-seq data for effector cytokines and anti-viral mediators.   

We thank the reviewer for this constructive comment. The scRNA data also indicated the low expression level of IFN-γ. According to the requirements, we have added the comparison of the levels of GrB and CD107a in the conventional T cells and DP T cells. We have also added a heatmap of sc-RNA data reflecting the mRNA level of effector cytokines and anti-viral mediators in DP T cells in Fig 4F.

7)    Do the authors know if the DP population present in HFRS patients is specific for Hantaan antigens?  Alternatively, is this a population of bystander activated cells?  The authors should discuss this possibility.  It is perhaps beyond the scope of this study but a 24hr restimulation of PBMCs from HFRS patients with recombinant Hantaan virus NP or G1 proteins, followed by flow cytometry analysis of activation induced markers such as OX40, PD-L1, CD25 on T cells, or cytokine bead array analysis of culture supernatants might reveal antigen specificity.  Something similar was done by Van Epps, Schmaljohn and Ennis in 1999. Van Epps HL, Schmaljohn CS, Ennis FA. Human memory cytotoxic T-lymphocyte (CTL) responses to Hantaan virus infection: identification of virus-specific and cross-reactive CD8(+) CTL epitopes on nucleocapsid protein. J Virol. 1999 Jul;73(7):5301-8. doi: 10.1128/JVI.73.7.5301-5308.1999. PMID: 10364276; PMCID: PMC112585.

We appreciated very much for your professional suggestions. We mentioned it in our discussion section that we did not explore the antigen specificity of DP T cells in this study. The staining of tetramer loaded with peptide specific to HTNV can also test the antigen specificity of DP T cells. However, the staining of tetramer needs to screen out the HFRS patients with proper MHC haplotype typing firstly. But it is not in the epidemic season at present. We cannot collect enough HFRS patients whole blood samples right now. This assay will be performed in our future study.

We are working on the bystander activated T cells at present. The manuscript has been submitted recently. However, we have no idea whether the DP T cells can be activated through bystander pathway. This is a very good question. We discussed the possibility in the discussion section.

8)    In the discussion, the authors mention that in vitro HTNV infection did not alter the percentage of DP cells.  However, the authors don’t demonstrate that the cells in culture were even infected with virus.  Is there any evidence that any of the cells in culture were infected with HTNV?  As is, Supplementary Figure 4 is an uninterpretable negative result.  Reference 12 provided by the authors has a nice example of demonstrating HTNV infection using a NTHV-NP antibody and is presumably why this experiment was performed.  This paper should be referenced in this portion of the discussion as it is very relevant.   If cells, particularly monocytes of B cells are infected during this assay, is there evidence that the DP or conventional single positive T cell populations respond to the infected monocytes or B cells?  Again, this would demonstrate antigen specificity.    

Thank you for these very constructive comments. We performed in vitro infection and demonstrated that the DP T cells cannot be infected in vitro with HTNV successfully, using a NTHV-NP antibody published by Liu R, et al. As previously reports, HTNV can infected monocytes. Whether the monocytes or B cells can present the HTNV antigen to DP T cells is unknown. Our further studies are needed to address this problem.

9)    In the abstract, the authors mention that, “HTNV infection caused the upregulation of DP T cells in the peripheral blood, which were correlated with disease severity”.  However, in the text for Figure 1, the authors state that, “although the mild/moderate HFRS patients had higher levels of DP T cells than severe/critical HFRS patients, there was no significant difference between different severity groups”.  These two statements do not agree and need to be addressed.

Thank you for pointing out our mistake. We apologize to have overlooked this. We have revised “HTNV infection caused the upregulation of DP T cells in the peripheral blood, which were correlated with disease severity” to “HTNV infection caused the upregulation of DP T cells in the peripheral blood, which were correlated with disease stage”.

10)   The authors suggest that DP T cells in HFRS might be involved in the pathogenesis of hemorrhage and renal failure.  Please discuss how these cells might contribute to pathogenesis.  Again, including  RNS-seq data for effector cytokines and anti-viral mediators  for the DP population would be incredibly valuable and should be included.

Many thanks for your good suggestions. The scRNA-seq data for effector cytokines and anti-viral mediators were added in Figure 4 in the latest version. Based on these, we thought that DP T may contribute to the pathogenesis by expression Granzymes. These mediators are thought to disrupt the endothelial cell barrier by cytotoxicity. We have added this point in our discussion.

11)   Please discuss what, if anything, is known about the presence of DP T Cells in hantavirus disease caused by other hantaviruses (ie Puumala, Sin Nombre, Andes).

We thank the reviewer for this very constructive suggestion. As we know, this is the first study on the DP T cells in hantavirus disease. We have rewritten the concluding paragraph to emphasize the point.

Round 2

Reviewer 1 Report

The authors have made the changes and modifications that make the manuscript publishable.

Author Response

We thank the reviewer for the positive evaluation of our manuscript.

Reviewer 3 Report

Thank you.  Your manuscript is greatly improved both scientifically and grammatically.  Please find below a few very minor grammatical corrections to address otherwise I have no further comments for the authors.   

Page 2.  Just after reference [17].  Please change from “Under pathological states, the number of DP T cells elevated in different diseases” to “Under pathological states, the number of DP T cells can become elevated in different diseases”.

Page 5.  Section 2.3 Flow Cytometry.  Second paragraph.  Please change from, “The antibody specific to CD107A was added when simulation” to “The antibody specific to CD107A was added during PBMC simulation”.

Page 6.  Section 2.3 Flow Cytometry.  Please change from, “All the antibodies used in flow cytometry is summarized in Supplementary Table 1” to “All the antibodies used in flow cytometry are summarized in Supplementary Table 1”.

Page 6.  Section 2.4  In vitro infection.  Please change from, “The nucleocapsid protein (NP) of HTNV in DP T cells were detected by flow cytometry…” to “The nucleocapsid protein (NP) of HTNV in DP T cells was detected by flow cytometry…”

Page 10.  Second paragraph.  Please change from, “In normal controls, it is clearly to see that the DP T cells were…” to “In normal controls, it is clear to see that the DP T cells were…”

Page 14.  Second paragraph.  Please change from, “After comparing the production of GrB and CD107a between DP T cells and conventional T cells, we noticed that DP T cells exhibited similar cytokine production profile with CD8+T cells…”  to “After comparing the production of GrB and CD107a between DP T cells and conventional T cells, we noticed that DP T cells exhibited a similar cytokine production profile with CD8+T cells…”

Page 19.  First full paragraph.  Please change from, “In our study, we gated CD3 molecule at first place.” to “In our study, we gated on the CD3 molecule first.” 

Author Response

We thank the reviewer to give us these suggestions to improve this manuscript. They are very helpful. We have replied the reviewers’ comments point by point. We hope it could meet the criteria for publication this time.

  1. Page 2.  Just after reference [17].  Please change from “Under pathological states, the number of DP T cells elevated in different diseases” to “Under pathological states, the number of DP T cells can become elevated in different diseases”.

We appreciate very much for your careful review. We have made modifications according to your requirements.

  1. Page 5.  Section 2.3 Flow Cytometry.  Second paragraph.  Please change from, “The antibody specific to CD107A was added when simulation” to “The antibody specific to CD107A was added during PBMC simulation”.

We appreciate very much for this suggestion. We have revised the sentence.

  1. Page 6.  Section 2.3 Flow Cytometry.  Please change from, “All the antibodies used in flow cytometry is summarized in Supplementary Table 1” to “All the antibodies used in flow cytometry are summarized in Supplementary Table 1”.

We thank the reviewer to point out this important issue. We have revised “is” to “are”.

  1. Page 6.  Section 2.4  In vitro infection.  Please change from, “The nucleocapsid protein (NP) of HTNV in DP T cells were detected by flow cytometry…” to “The nucleocapsid protein (NP) of HTNV in DP T cells was detected by flow cytometry…”

Thanks for pointing out this mistake. We have revised “were” to “was”.

  1. Page 10.  Second paragraph.  Please change from, “In normal controls, it is clearly to see that the DP T cells were…” to “In normal controls, it is clear to see that the DP T cells were…”

We appreciate very much for your careful review. We have revised “clearly” to “clear”.

  1. Page 14.  Second paragraph.  Please change from, “After comparing the production of GrB and CD107a between DP T cells and conventional T cells, we noticed that DP T cells exhibited similar cytokine production profile with CD8+T cells…”  to “After comparing the production of GrB and CD107a between DP T cells and conventional T cells, we noticed that DP T cells exhibited a similar cytokine production profile with CD8+T cells…”

 Thanks for pointing out this mistake. We have revised the sentence in the revised version of the manuscript.

  1. Page 19.  First full paragraph.  Please change from, “In our study, we gated CD3 molecule at first place.” to “In our study, we gated on the CD3 molecule first.” 

Many thanks for your good suggestions. We have revised the “In our study, we gated CD3 molecule at first place.” to “In our study, we gated on the CD3 molecule first.” in the revised manuscript in red.
